# The Revival of the Battle between David and Goliath in the Enteric Viruses and Microbiota Struggle: Potential Implication for Celiac Disease

**DOI:** 10.3390/microorganisms7060173

**Published:** 2019-06-14

**Authors:** Aaron Lerner, Ajay Ramesh, Torsten Matthias

**Affiliations:** AESKU.KIPP Institute, Mikroforum Ring 2, 55234 Wendelsheim, Germany; ramesh@aesku.com (A.R.); matthias@aesku.com (T.M.)

**Keywords:** virus, bacteria, virome, phageome, microbiome, gut, intestinal lumen, horizontal gene transfer, celiac disease

## Abstract

The human gut is inhabited by overcrowded prokaryotic communities, a major component of which is the virome, comprised of viruses, bacteriophages, archaea, eukaryotes and bacteria. The virome is required for luminal homeostasis and, by their lytic or synergic capacities, they can regulate the microbial community structure and activity. Dysbiosis is associated with numerous chronic human diseases. Since the virome can impact microbial genetics and behavior, understanding its biology, composition, cellular cycle, regulation, mode of action and potential beneficial or hostile activities can change the present paradigm of the cross-talks in the luminal gut compartment. Celiac disease is a frequent autoimmune disease in which viruses can play a role in disease development. Based on the current knowledge on the enteric virome, in relation to celiac disease pathophysiological evolvement, the current review summarizes the potential interphases between the two. Exploring and understanding the role of the enteric virome in gluten-dependent enteropathy might bring new therapeutic strategies to change the luminal eco-event for the patient’s benefit.

## 1. Introduction

Several human chronic disease incidences are increasing in the last few decades—cancer, allergy and autoimmune diseases are some of these [1]—and all have proven genetic backgrounds, but they are induced and influenced by the changing environment. The arguments about the differential influences of the genes/environment are endlessly expanding. Since genetic, inter-generation, eukaryotic vertical changes are long-term events and the environment is changing constantly, intra generationally and epigenetical environmental changes may play the major role in disease development [2]. A major environmental factor that exerts a driving force on human health are prokaryotes [3,4,5,6]. It appears that the human enteric compartment is an ideal niche for the coevolution of numerous prokaryotes cohabiting eukaryotic guts in a bi-directional symbiotic environment. In fact, human evolution started billions of years after the unicellular creatures appeared on earth and they diversified to colonize all of Earth’s habitable environments before invading the human gut [7]. A plethora of information is available on the intestinal microbiome/dysbiome composition and diversity in human homeostasis and morbidity. However, not much is known on the enteric viruses. In this regard, the present review will concentrate on the human enteric virome, its cross-talks with the microbiome and its potential importance in autoimmune diseases, with celiac disease (CD) taken as an example.

### 1.1. The Microbiota–Dysbiota–Mobilome Networks

Single-cell microorganisms shaped our planet, multicellular organisms and the human gut ecology. Microbiome/dysbiome compositional and diversity balances change across the human lifespan and are associated with specific chronic diseases, although, no causality has, as yet, been established [8,9]. Increasing information is being accumulated, presenting intestinal luminal eco-events as prime sites in a melting pot with irradiating consequences to remote organs [10]. Gut axes to the brain, liver, kidney, thyroid, heart, bone, joints, skin, or even to stomach or gut itself, for example, were suggested [10,11,12,13,14]. Playing a crucial role, the luminal microbial communities consume, tie and bind the nutrients, induce enzymatic and metabolic events, secrete mobilome, transmit foreign genetic cargo, and change epigenetic phenomenon, thus affecting the evolutional struggle between bugs and humans [8,10,15,16,17]. Certain gut microbes can degrade compounds such as drugs, providing a mechanism by which they may regulate a multitude of aspects of health [18]. In this regard, the gut microbiota impacts a plethora of animal or human diseases, such as neuroimmune and neurodegenerative, enteric and central, motility, behavioral, cardiovascular and metabolic diseases [19]. The microbial mobilome that modulates or affects the microbiome–gut–brain axis is just starting to unravel.

In fact, intestinal microbes were shown to secrete and/or consume a plethora of neurotransmitters, and norepinephrine, dopamine, gamma-aminobutyric acid (GABA), and serotonin are some of them [11,19]. More so, preliminary studies indicate that enteric microbiota manipulations can impact neurotransmitter levels [19,20].

Intriguingly, viruses and bacteriophages are now being recognized as abundant and diverse gut community members. They might orchestrate the above-mentioned human compartmental occurrences and participate in luminal homeostasis or abnormalities.

### 1.2. The Enteric Viruses, The Phageome

The intestinal virome is part of the microbiome and includes members that target human cells and microorganisms, including archaea, fungi, protozoa and bacteria—the best studied of which are human viruses and those that infect cultivable bacteria. However, cultivation-independent sequencing methods are developing and bring to light many bacteriophage, probably mostly double-stranded DNA varieties. Much less studied are single-stranded and RNA viruses, although increasing transcriptome sequencing will likely uncover many more new types.

Numerous publications exist on the human microbiome and the place of the corresponding dysbiota in specific human chronic conditions. However, the role of the gut virome has just started to be considered. Noteworthy is the fact that the gut viruses outnumber microbes in a ratio of 10:1 [21]. Their main impacts occurring by predation of the microbiota and via integration into the host genome [22,23,24]. The fate of the infected microbiota can be either cell death by lysis or transient symbiosis, called lysogeny, resulting in chronic infection [22,24]. The stable, balanced and homeostatic microbiome cannot maintain itself without the enteric phageome (collection of bacteriophages). The core and common prophages and bacteriophages are globally distributed and comprise the healthy gut phageome that is crucial in maintaining normal microbiotic diversity and functionality [25]. Interestingly, as dysbiosis is related to specific intestinal and extra-intestinal entities [8], phageomic aberrations are, similarly related to human diseases. In ulcerative colitis and Crohn’s diseases, core phages are reduced [25], while an increase was recently observed in type 2 diabetes [26].

In reality, the intestinal lumen can be considered as an optimal environment for microbial community interactions and prokaryotic–eukaryotic cell-to-cell cross talk. The high cell density, continuous inflow of nutrients (including proteins and metabolites), stable temperature, moisture, conjugation opportunities and biofilm structures set the stage for viral evolution, predation, replication and survival [17]. The local virome biodiversity changes along the human life cycle. The phage load decreases, while the abundance and complexity of the microbial populations increases substantially during aging [22]. It seems that intestinal bacterial composition and diversification occurs at the expense of the virome communities [27]. Several endogenous and environmental factors can affect virome composition and diversity. Prophage reservoir, enteric microbial community, human host immune systems, life cycle, geography and nutrition are influential [22,23,24].

Due to next-generation sequencing technologies and sophisticated bioinformatics tools, the metagenomic sequence data of the virome is expanding, but the gene functions are lagging behind. The metagenome of human feces yielded an estimate of 2.5–6.5% for horizontal gene transfer, analyzed by compositional methods [28], and part of it is originates from viruses. No doubt that lateral gene transfer is a powerful means by which the virome can genetically restructure gut holobionts [17,28,29,30].

### 1.3. Virus–Microbiota Cross Talk

Being absolute parasites, the phageome has to penetrate living cells, be it pro or eukaryotes. The virome–microbiome cross interactions regulate the lumen gut communities, shaping their taxonomy and functional diversity and impact microbial cell evolution that can acquire or spread genetic cargo [17,23,24,25,26,27,28,29,30], thus impacting the homeostatic environment and avoiding its imbalance (Figure 1). By lysing, the phages promote the prokaryote evolution and biodiversity and ameliorate the resource usage of the intestinal dwellers. Further, lysis can release cell constituents such as LPS that can trigger immune response by activating their binding receptors [31]. By lysogeny, however, the prophage carrier microbes can acquire beneficial genes for evolution, survival and overcoming competing species. Acquiring toxic or hostile factors might help the lysogens to fit into the luminal compartment and execute their biological functions. Finally, host substrate utilization can be reduced [24]. Figure 2 schematically describes the lysogenic and lytic cycles, and their consequences. The bidirectional relations between the microbiome and the virome, or more specifically, between the enteral viruses and the microbial host was revolutionized by the discovery of the Clustered Regulatory Interspaced Short Palindromic Repeats (CRISPR). It is a multi-functional, evolutionary evolving bacterial and archaeal memory immune mechanism that defends prokaryotic cells against invaders such as phages, plasmids or other mobile genetic sequences [32,33]. There is never a dull moment in the highly populated gut lumen, where the virome with its pathogenic or beneficial capacities needs to confront the antiviral machinery, thus, boosting microbiome robustness. Unsurprisingly, recent research has uncovered a variety of mechanisms by which phages can fight back, including via the acquisition of their own CRISPR-Cas systems and various anti-CRISPR systems [34]. Monozygotic twins are a human genetic model to study the interaction between the intestinal phageome and the microbiome [35]. When microbiome–discordant twins were compared to concordant ones, the authors found that a more dissimilar virome was lodged in the discordant ones and the richer is the microbiome, the richer is the corresponding phageome. The phenomenon was driven by bacteriophages and not by eukaryotic viruses [35]. Studying gut *bifidobacteria*, a bacterium known to be associated with CD, and its relation to phages, it was revealed that the incidence of integrated prophages in *Bifidobacterium* genome is quite substantial [36]. This opens a window to the potential modulatory role of those synergic phages on the human gut-associated *bifidobacteria* populations. Finally, a totally unexplored topic is the bacteriophages repertoire associated with probiotics. It might represent an additional facet to the multiple adverse effects that were recently described [37]. Finally, bacteria facilitate enteric virus co-infection of mammalian cells, thus promoting the genetic recombination between viruses and restoring intestinal phageome fitness [38]. It can be concluded that the luminal gut virus–microbe cross relations are bidirectional and are part of the luminal compartment homeostatic balance.

## 2. Viruses That Might Be Involved in Celiac Disease Development

The exploration of infections as an environmental trigger for celiac disease (CD) initiation or progression experienced highs and lows [39,40]. With regards to viruses, several were suggested as causative agents, although this remains debatable. The adenovirus 12 E1A was mentioned as involved in CD pathogenesis [41] but was disqualified later on [42]. Some advocated the role of rotavirus [43], but the association was denied [44]. Hepatitis C and enteroviruses were also suggested, but not proved to be important [39,40]. The latest suggested viruses to be involved in CD development are reoviruses [45]. In fact, in a mice model, viral infection induced a break of oral tolerance to dietary proteins, promoted a proinflammatory phenotype in dendritic cells, resulting in pathogenic T cell response [45]. Higher anti-reovirus serological reactivity was shown in CD patients, thus, supporting the role of infection with an innocuous virus in triggering CD development. Future studies need to be performed to answer the question: Does CD “go viral”? [46].

## 3. Celiac Disease in Short

Celiac disease is a frequent, life-long autoimmune disease mainly targeting the proximal small intestine with multiple systemic extra intestinal manifestations, in genetically susceptible individuals [47]. It affects 1–1.5% of the Western world and its epidemiology is changing constantly. In recent decades, the typical presentation of infant chronic diarrhea, protuberant abdomen, weight loss and malabsorption is replaced by extra intestinal presentations at a higher age. Gluten is, so far, its only causative offending environmental factor and withdrawal of wheat, barley, rye and oat containing nutrients is its only approved therapy. The endogenous enzyme, tissue transglutaminase (tTG), is the autoantigen of the disease and IgA-anti tTG is the most frequently used diagnostic serological marker [48]. Lately, a new serological marker, targeting neoepitope complexes formed when gliadin docks the tTG enzyme was described [15]. More recently, a microbial enzyme, namely, microbial transglutaminase (mTG), a heavily used industrial processed food additive, that functionally imitates the tTG, was shown to induce specific antibodies in CD patients [49]. Potentially, it might represent a new environmental factor that drives CD autoimmunogenesis [50]. Genetically, the *HLA-DQ2* and *HLA-DQ8* genes are the most important, predisposing factors [48].

The pathogenesis of the disease is dominated by the interplay between the intestinal innate and reactive immune systems, resulting in gut inflammation and destruction. In fact, each pathogenic step in the CD mucosal destructive cascade forms the basis for the development of future therapeutic strategies to replace the existing gluten-free dietary therapies [48].

## 4. The Microbiome Signature in Celiac Disease

Multiple studies exploring the celiac microbiome were summarized recently [48,51]. It is generally accepted that the altered microbiome/dysbiome balance may exist as a luminal modifier of CD evolvement. Many reports described the decrease in *Bifidobacterium* and *Lactobacillus* species, while the proportions of *Bacteroides* and *Proteobacteria* were increased. At least in a Swedish CD epidemic, *Clostridium* spp., *Prevotella* spp. and *Actinomyces* spp. were identified in the proximal small bowel in the patients. More recently, other microbes that were suggested to be associated with the disease were: *Campylobacter jejuni*, virulent *Escherichia coli*, *Staphylococcus* spp., *Bacteroides fragilis* and *Neisseria flavescens* [51]. Despite the fact that the dysbiosis is established in CD, the causative factors and their mechanisms are far from being elucidated. For now, CD and its enteric dysbiosis are associative and no cause and effect relationship was determined.

## 5. Potential Virome Impacts on Celiac Disease Intestinal Eco-Events

Based on the current knowledge on CD environmentally induced pathogenesis and the latest association of the enteric virome affecting luminal prokaryotic homeostasis, there is a strong logical background to explore the intestinal phageome in CD development. Table 1 summarizes the pathophysiologic events or pathways by which gut phageome might play a role in CD genesis.

Several additional mechanisms might be induced or enhanced by phages in CD genesis. Alternatively, phage therapy may represent a new therapeutic strategy for the treatment of this autoimmune condition.
The gut phageome may select microbes that do not digest gluten or lack the glutenase capacities, thus increasing the luminal gluten load to the chagrin of the affected patients.Bacteriophages are used in the nutritional and processed food industries, representing an additional phageomic cargo that enters the gut lumen [52,53], potentially changing the viral–bacterial equilibrium.Bacteriophages or prophages can increase microbiome antibiotic resistance [62,63,64,65], which could be disadvantageous for CD patients.Lytic and synergic enteric phage treatment modulated the composition and diversity of the microbiome. Lysis promoted a beneficial and equilibrated luminal ecosystem, while the temperate phage can promote conditions enabling pathogenic conditions, at least as shown in a mice model [66]. Can phage therapy reverse the dysbiosis associated with CD?Lytic bacteriophage are suggested to control multidrug or antibiotic resistance and pathogenic invasive bacteria. The strategy might help to fight enteric infections or gut-originated autoimmune diseases, as was suggested for Crohn’s disease [67,68].Recently, the mTG, a heavily used processed food additive, capable of cross-linking proteins, including gliadins, was suggested to enhance CD development [15,49,50,69]. The enzyme is considered as a bacterial virulence factor [69] and one wonders whether bacteriophage therapy could reduce its bacterial-originated luminal enzymatic burden, potentially benefiting the patients. A very interesting topic is the virus mTG activity. Long ago, the group of Prof. Aravind L. at NIH detected an ancient core transglutaminase fold in prokaryotic enzymes [70,71]. Transglutaminase activity is important to *Candida albicans* and *Saccharomyces cerevisiae* function and survival [72,73]. Both were suggested to impact CD evolvement [74,75]. Interestingly, large viruses [76] and the recently explored megaphages [77] might harbor transglutaminase-like sequences (Jillian F. Banfield, personal communication). Intriguingly, those *Prevotella*-infected megaphages occur in the human gut microbiome, and *Prevotella jejuni* spp. is one of the enteric bacteria associated with naive CD [48,78]. Could the mTG viral cargo impact the immunogenicity or the potential pathogenicity of the enzyme to post translate and modify gluten, making it less tolerant and more toxic to CD patients?Due to the emergence of viruses as triggers of CD [45,46,79] and the discovery of the CRISPR machinery [32,33], anti-viral protective memory can protect against or counteract the suggested CD-inducing viruses. Phages evolve their genomes to evade immunity. Several examples were described. One of them is the phage genome evolution in *Streptococcus thermophilus*, driven by CRISPR immunity [80]. The impact of the anti-viral CRISPR protective apparatus on the intestinal CD phageome is far from being elucidated.Most recently, new light was shed on the role of bacteriophages in aggravating enteric inflammation and colitis [81]. Gogokhia et al. reported novel mechanistic pathways whereby the bacteriophages are operating. The phages activate interferon γ-mediated immune responses via *TLR9* and exacerbate colitis in mice and their increased abundance in ulcerative colitis patients is correlated with mucosal interferon responses. Since the present review deals with CD, those observations on colitis might also apply to CD intestinal inflammation. The pro-inflammatory interferon γ is involved in mucosal damage, dictates epigenetic immune outcomes and is sensitive to gluten challenge in CD patients [82,83,84,85,86]. Interestingly, mucosal *TLR9* gene expression is increased in CD, suggesting the contribution of the microbiota or dysregulation of the immune response to the small bowel flora in the CD intestine [87,88]. The bacteriophage’s involvement in intestinal inflammation in colitis can potentially take place in the CD gut mucosal compartment.


## 6. Conclusions

Increasing knowledge is accumulating on the importance of the gut virome, especially gut-associated bacteriophage, in health and disease. Viruses are an integral part of biological diversity. They are important in maintaining luminal homeostasis and their lytic or synergic activities clearly shape gut microbiome function. Microbiome biodiversity is impacted by them throughout the human life cycle, under normal and abnormal conditions. They may confer beneficial functions or detrimental activities. Exploring their composition, diversity, lateral genetic transmission, mechanistic pathways and regulation might bring new therapeutic strategies to change the interphase between bugs and humans for human benefits. One wonders whether the recent review on the “Human Gut Microbiome—A Potential Controller of Wellness and Disease” [89] will induce studies on the enteric phages’ place as an additional controller of the human intestinal ecology [90]. Alternatively, could antiviral vaccinations open a new therapeutic strategy to fight autoimmunity as most recently suggested in CD by rotavirus vaccination [91]?

The role of the intestinal virome in human chronic disease is just starting to unravel, but their role in CD development remains obscure. This review highlights the potential interfaces between the gut virome and CD pathogenesis with the goal of encouraging the scientific community to explore the topic to provide pathways to renew the physiologically beneficial microbiome, lyse the offending microbes or increase their resilience to hostile viruses. The biblical battle between David and Goliath was between unequal men, where the smaller, but the more sophisticated one won. The virus is smaller compared to the bacteria, both competing on the same overpopulated luminal niche. Respectfully, the biblical battle between David and Goliath resembles the phageome–microbiome relationship, where the future will tell us who is more sophisticated and dominates the arena.

## Figures and Tables

**Figure 1 microorganisms-07-00173-f001:**
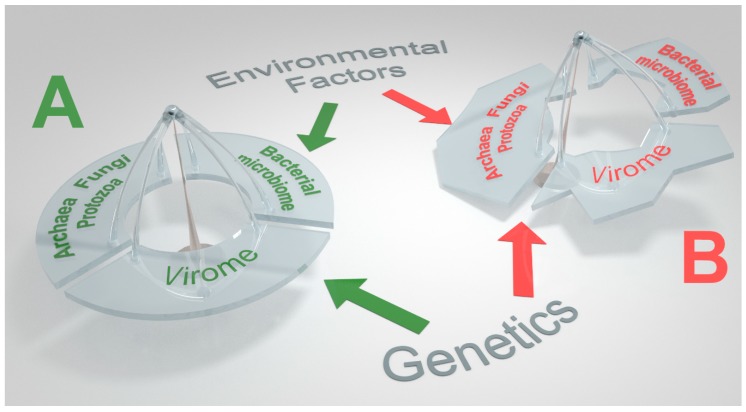
A schematic presentation of the intestinal prokaryotic interrelationship during homeostatic (**A**) or imbalanced states (**B**).

**Figure 2 microorganisms-07-00173-f002:**
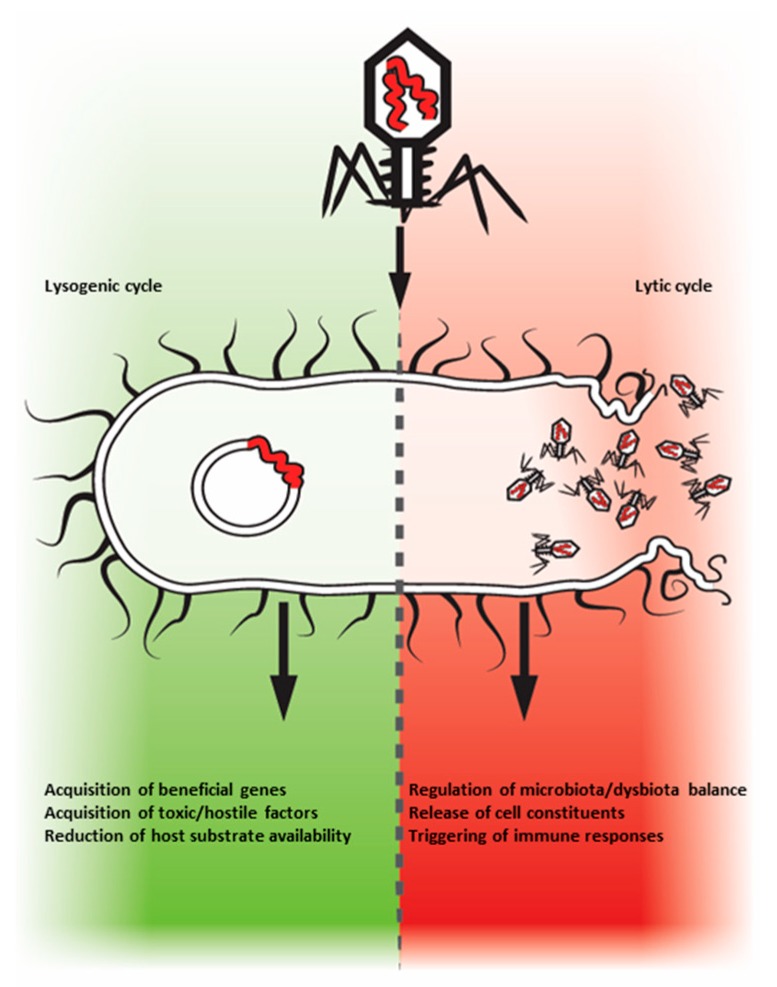
The lysogenic and lytic cycles of the enteric phageome and their local and systemic consequences.

**Table 1 microorganisms-07-00173-t001:** Potential enteric virome effects on the intestinal and systemic effect in CD development.

Potential Enteric Virome Role	Potential Effect in CD	Reference
Regulators of microbial diversity and composition	Maintenance of altered enteric microbiota	[24]
Affecting microbiota/dysbiota ratio	Decreased balance	[23,24]
Horizontal transfer of genetic mobile elements	Increase microbial hostility	[17,22,23,24,25,26,27,28,29,30]
Providing lysed bacterial components	Enhanced local /systemic inflammation	[24,31]
Viral genetic and metabolic mobilome with potential systemic effects	Pro-inflammatory factors and state	[17,22,23]
Boosting the altered microbiome robustness.	Induction/maintenance of intestinal dysbiosis	[32,33]
Reduction of microbial substrate utilization	Induction of stressful environment	[24]
Induction of life cycle alteration in microbiome composition and biodiversity	Age-dependent microbiome abnormalities	[22,24]
Specific lysis of *Lactococcus* and *Bifidobacterium* spp.	Decreased diversity of physiologic microbiota	[52,53,54,55,56,57,58,59,60,61]

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
