# Peer review of "The Revival of the Battle between David and Goliath in the Enteric Viruses and Microbiota Struggle: Potential Implication for Celiac Disease"

_microorganisms, 2019, doi:10.3390/microorganisms7060173_

Round 1
Reviewer 1 Report
Major revisions
Authors wrote this review with the aim to discuss about the association with dysbiosis and celiac disease. This is stated in the title, however, the first part of the review seems to be larger than the parte dedicated to celiac disease itself. A very poor background is present about celiac disease and it should be improved.
I found inconsistent the numbered list which starts at page 5 line 172. It shortens the discussion and this is not useful to the readers. It would be better to talk over every point in a more conversational way.
The reference to David and Goliath war in the title seems to be sonorous and doesn’t have a real basis as far as can be found in the text.
Self-citations seem to be excessive (Lerner A. is cited from reference 17 to reference 26 and also further, alone or with Matthias T). Authors should avoid too much self-citations.
Minor revisions
Page 3, line 126 please correct by adding ‘’celiac disease (CD)’’ instead of ‘’CD’’, and consequently delete ‘’celiac disease (CD)’’ at Page 4, line 139 and use only ‘’CD’’
Some typing errors must be revised through all the manuscript.
Author Response
Reviewer #1
We appreciate greatly the comments of Reviewer #1. They were very instructive. Thanks.
Here are our responses: (highlighted in yellow)
1. Authors wrote this review with the aim to discuss about the association with dysbiosis and celiac disease. This is stated in the title, however, the first part of the review seems to be larger than the parte dedicated to celiac disease itself. A very poor background is present about celiac disease and it should be improved. A whole paragraph ; “celiac disease in short” was added, lines 154-172. References 47-50 were added.
I found inconsistent the numbered list which starts at page 5 line 172. It shortens the discussion and this is not useful to the readers. It would be better to talk over every point in a more conversational way. “celiac disease in short” clarifies some of the numbered list. The list was expanded. Lines 194-238 and clarifing comments were added
The reference to David and Goliath war in the title seems to be sonorous and doesn’t have a real basis as far as can be found in the text. An explanation was added in page 7, lines 256-260.
Self-citations seem to be excessive (Lerner A. is cited from reference 17 to reference 26 and also further, alone or with Matthias T). Authors should avoid too much self-citations. Ten self-references were deleted.
Minor revisions
Page 3, line 126 please correct by adding ‘’celiac disease (CD)’’ instead of ‘’CD’’, and consequently delete ‘’celiac disease (CD)’’ at Page 4, line 139 and use only ‘’CD’’ Done
Some typing errors must be revised through all the manuscript. Revised by a native speaking scientist.
Reviewer 2 Report
Figure 1 would benefit if it was presented as a regular drawing using a graphic tool, for example a Power-point.
Author Response
Reviewer #2
Figure 1 would benefit if it was presented as a regular drawing using a graphic tool, for example a Power-point. Thanks for the comment. Figure 1 was changed, simplified and clarified by adding (A) and (B).
Reviewer 3 Report
I had just viewed the NIH Wednesday Afternoon Lecture given by Julie Pfeiffer (Getting by with a little help from their friends: How bacteria aid virus infection) when I was asked to be a reviewer. This prepared me for your excellent paper. Thank you for your references which I will use for my summer reading.
Erickson AK, Jesudhasan PR, Mayer MJ, Narbad A, Winter SE, Pfeiffer JK. Bacteria Facilitate Enteric Virus Co-infection of Mammalian Cells and Promote Genetic Recombination. Cell Host Microbe. 2017;23(1):77–88.e5. doi:10.1016/j.chom.2017.11.007
Author Response
Reviewer #3
I had just viewed the NIH Wednesday Afternoon Lecture given by Julie Pfeiffer (Getting by with a little help from their friends: How bacteria aid virus infection) when I was asked to be a reviewer. This prepared me for your excellent paper. Thank you for your references which I will use for my summer reading.
Erickson AK, Jesudhasan PR, Mayer MJ, Narbad A, Winter SE, Pfeiffer JK. Bacteria Facilitate Enteric Virus Co-infection of Mammalian Cells and Promote Genetic Recombination. Cell Host Microbe. 2017;23(1):77–88.e5. doi:10.1016/j.chom.2017.11.007
Thanks for the lovely complements. The paper was imbeded in the text. Reference No 38.
Round 2
Reviewer 1 Report
Authors addressed completely all the issue.